# Synthesis and Application of Liquid Metal Based-2D Nanomaterials: A Perspective View for Sustainable Energy

**DOI:** 10.3390/molecules28020524

**Published:** 2023-01-05

**Authors:** Gengcheng Liao, Long Ren, Zixuan Guo, Hui Qiao, Zongyu Huang, Ziyu Wang, Xiang Qi

**Affiliations:** 1Hunan Key Laboratory for Micro-Nano Energy Materials and Devices, School of Physics and Optoelectronic, Xiangtan University, Xiangtan 411105, China; 2State Key Laboratory of Advanced Technology for Materials Synthesis and Processing, International School of Materials Science and Engineering, Wuhan University of Technology, Wuhan 430070, China; 3Hunan Key Laboratory of Two-Dimensional Materials, Hunan University, Changsha 410082, China; 4Suzhou Institute of Wuhan University, Suzhou 215125, China

**Keywords:** two-dimensional metal oxide materials, liquid metal, energy reduction

## Abstract

With the continuous exploration of low-dimensional nanomaterials, two dimensional metal oxides (2DMOs) has been received great interest. However, their further development is limited by the high cost in the preparation process and the unstable states caused by the polarization of surface chemical bonds. Recently, obtaining mental oxides via liquid metals have been considered a surprising method for obtaining 2DMOs. Therefore, how to scientifically choose different preparation methods to obtain 2DMOs applying in different application scenarios is an ongoing process worth discussing. This review will provide some new opportunities for the rational design of 2DMOs based on liquid metals. Firstly, the surface oxidation process and in situ electrical replacement reaction process of liquid metals are introduced in detail, which provides theoretical basis for realizing functional 2DMOs. Secondly, by simple sticking method, gas injection method and ultrasonic method, 2DMOs can be obtained from liquid metal, the characteristics of each method are introduced in detail. Then, this review provides some prospective new ideas for 2DMOs in other energy-related applications such as photodegradation, CO_2_ reduction and battery applications. Finally, the present challenges and future development prospects of 2DMOs applied in liquid metals are presented.

## 1. Introduction

Growing energy shortages and environmental problems have been the focus of the world’s attention in recent years. There is a commitment to develop effective strategies for renewable energy and environmental remediation due to these challenges today. In recent years, utilizing the unique advantages of low-dimensional semiconductor nanomaterials has received extensive attention in the energy field [1,2,3,4,5,6,7]. Among many common 2D materials, two dimensional metal oxides (2DMOs) have been further explored due to its advantages and have received extensive attention in energy applications, such as catalysis [8,9], battery [10,11,12] and CO_2_ reduction [13,14]. For example, 2DMOs have long been considered as excellent potential electrode materials for lithium batteries, not only they are easy to fabricate on a large scale, but also they keep in compatibility of achieving abundant redox reactions and involving different ions [15,16], which helps to improve specific capacity/capacitance. In the field of CO_2_ reduction, 2DMOs are considered to be a good candidate to promote CO_2_ reduction reaction, which is due to an electrochemical process that could reduce the active metal phase [17,18], resulting in many grain boundaries leading to a delighted result that could improve the selectivity of CO_2_ reduction reaction. In the field of photodegradation, compared with traditional TiO_2_ catalysts, some metal oxides can exhibit more suitable redox potential [19], leading to higher driving forces for photocatalysis. However, 2DMOs are considered to be non-lamellar materials, compared with the common layered materials, the force between the bond energies is very weak [20,21,22,23,24,25,26,27], which is easy to cause the dangling bond, and enables the material surface with more chemical activity. Thus, it is not suitable to obtain two-dimensional metal oxide materials by common methods [28,29].

In recent years, liquid metals have received extensive attention due to their high electrical conductivity, thermal conductivity, and environmental friendliness, and have been used in applications such as bionic robots, flexible optoelectronic devices, and medical bioskeletons [30,31,32,33,34,35]. Surprisingly, latest researches reveal a new concept that liquid metals have been unfolded great possibility of fabricating 2DMOs due to its good solubility [36], chemical reactivity [37], weak interfacial force [38], and liquid solid phase transition [39]. Specifically, low melting point liquid metals, such as gallium (Ga), eutectic gallium–indium (EGaIn), Gallium-indium-tin (EGaInSn) et al. will form self-limiting oxide films in air at room temperature, which is considered a natural two-dimensional semiconductor. Affected by this point of view, it is undoubtedly a good way to obtain 2DMOs through liquid metal. It is worth noting that liquid metals, due to their attractive low melting points and chemical compatibility, have been used as templates and reaction platforms to provide metal oxide films and modified metal oxide films, respectively [40,41]. For nearly 10 years, parts of remarkable papers have been proposed and discussed as concluded in Figure 1. In this case, how to intelligently select different liquid metals oxide from those metals remain something to think about. However, for the better put into application, a single metal oxide cannot meet the broad application prospects of 2DMOs. Liquid metal alloying has received extensive attention due to the laws of thermodynamics. In detail, when two metals are sufficiently melted together, the metal with small Gibbs free energy preferentially generates the corresponding metal oxide on the surface of the alloy, changing the state of the liquid metal [42,43]. Nearly ultra-thin nano-scale oxide films can be obtained, and liquid metal particles wrapped in nano-scale oxide films can also be obtained. In this regard, compared with the traditional oxide acquisition method, it exhibits following advantages: (1) When the liquid metal is exposed to the air, an oxide layer will be continuously formed, and the collection method is simple and does not consume resources; (2) For a uniform displacement reaction, it can be designed different metal oxide films. By adjusting the parameters, their crystallization, size and thickness can be tuned by liquid metals; (3) Liquid metals can also act as participants, and their excellent high electrical and thermal conductivity, mobility and environmental protection properties can be used as charge transport layers to participate in lithium batteries, application scenarios related to supercapacitors.

This review focuses on low-dimensional nanomaterials on liquid metal surfaces, from the preparation of low-dimensional nanomaterials, including methods of LM-based materials preparation, and finally systematically reviewed in CO_2_ reduction, photodegradation and battery. In order to realize efficient environmental governance, develop clean energy and realize green ecology, this paper provides a possible solution. We then highlight future research directions and plans for further research.

## 2. Properties of Liquid Metal Based-2DMOs

### 2.1. Surface Oxidation

By changing the oxidation time and condition, dozens of metal oxides with different thicknesses and adjustable crystallinity can be effectively obtained from liquid metal [34,44,45]. Oxidation processes based on liquid metals can be divided into auto-oxidation processes [46] and polymetallic alloys-oxidation processes [47]. In detail, when it comes to an auto-oxidation process, when a metal is put at room temperature, its surface is exposed to air immediately will form a self-limiting oxide layer. In this process, the formation thickness of almost all metal oxides is related to the concentration of oxygen, the capacity of the oxygen content, the thickness of the oxide layer formed on the surface. Taking metal Ga as an example, Ga generally has two stable states, Ga^0^ and Ga^3+^ [48,49]. In the absence of oxygen, Ga^0^ state appears on the surface of Ga, which is regarded as an inherent metallic gallium state of liquid metal, while in the room temperature, Ga^3+^ state generates rapidly due to the oxide of liquid metal and coexists with Ga^0^ on the surface of gallium. Different from metal Ga, by adjusting oxygen content and oxidation reaction time, other liquid metals will appear in different oxidation forms and oxidation processes during the oxidation process. In the case of Sn, SnO and SnO_2_ are formed in the presence of oxygen (0–107 L) [50,51]. In natural environments, the high solubility of Sn in oxygen enhances the mobility of oxygen in Sn, allowing Sn to selectively form SnO on liquid metals. With the increase of oxygen content, the oxide layer becomes thicker, and the mixture of Sn_2_O_3_ and Sn_3_O_4_. And that’s because Sn is more likely to be attached to an oxygen group, which is related to the valence electrons of ^+2^ and ^+4^ valence electrons of Sn [52,53]. In fact, obtaining a single metal oxide from a liquid metal does not exhibit the advantages of LM. In this regard, polymetallic alloys-oxidation processes are considered to be a superior way to obtain selectable and functionalized 2DMOs. When two or more different metals are introduced, the new oxides formed on the surface of the alloy obey the laws of thermodynamics, which depend on their Gibbs free energy levels. In general, metal oxides with lower Gibbs free energy are preferentially formed on the alloy surface [54,55]. For example, under the same conditions, in the metal EGaIn, the Gibbs free energy of Ga_2_O_3_ is −998.3 KJ/mol, while the Gibbs free energy of InO is about 364.4 KJ/mol, which is much larger than the value of Ga_2_O_3_, that is to say, Ga_2_O_3_ will form on the surface of the alloy instead of InO [56,57]. It is worth noting that metals with higher Gibbs free energy may also form on the surface of the alloy, which is affected by temperature and proportion.

### 2.2. In-Situ Galvanic Replacement

Galvanic replacement has been extended to the liquid metal because it is easy to control and could be tuned for application of interest [37,58]. During this process, the metal is oxidized by another metal ion with a higher reduction potential. It is noteworthy that the liquid metal has received significant attention due to their excellent characteristic, thus the galvanic replacement has been widely used for the controlled fabrication of metal nanostructures towards gallium-based liquid metals. The low standard reduction potentials of Ga^3+^/Ga^0^, In^3+^/In^0^, AuBr_4_^−^/Au^0^, Ag^+^/Ag^0^, Sn^4+^/Sn^0^ (−0.529 V, −0.340 V, 0.854 V, 0.799 V, −0.138 V vs SHE, respectively) provide attractive driving forces for their electro-couple substitution reactions with various metal ions while the morphology of the products is determined by the concentrations of the metallic ions used in the solution during the galvanic replacement process [59,60,61]. When a GaInSn droplet in AuBr^4−^ aqueous solution, the standard reduction potentials are considered, which indicates that there will be a replacement reaction occurred due to a thermodynamic driving force. This redox reaction could be represented as: Ga^0^ + AuBr_4_^−^ = Ga^3+^ + Au^0^ + 4Br, Ga^0^ + 3Ag^+^ = Ga^3+^ + 3Ag^0^. This process can take place on the surface of a large number of liquid metals, even during sonochemical synthesis. On the basis of galvanic substitution, it is also possible to build metal/metal oxide structures around liquid metal particle (LMP), which not only improves stability of LMP under harsh conditions such as heavy oxidation and acid etching, but also regulates its essential properties such as thermal/electrical conductivity, catalytic activity, and SPR absorption [59,62]. This approach illustrates a new avenue of research ingalvanic replacement process that can also be applied to other systems to create the better materials such as liquid metal marbles or nanomaterials.

## 3. Synthesis of Liquid Metal-2DMOs

An atomic-level oxide film formed by self-oxidation of liquid metal at room temperature is considered as a new generation of two-dimensional semiconductor materials [52,56,63,64,65,66,67,68,69,70,71,72]. Unlike the common layered material exfoliation, the liquid metal core is a disordered structure made of metal bonds. When exposed to air, the surface of the liquid metal core will continuously form an oxide film. Notably, liquid metals have been applied into nanomaterials due to their electrical displacement reactivity and their high electrical conductivity, thermal conductivity, and environmental friendliness. In this section, this review will systematically summarize the methods of synthesizing liquid metal 2DMOs in recent years, classify them as liquid metal as a template (sticking method and gas injection method) and participants (ultrasound method) along with 2DMOs. In this case, some important factions of 2DMOs based on different liquid metal such as crystallization, thickness and ΔG are systematically summarized owing to the changes of condition, substrate or solvent of liquid metal in Table 1. At last, this section will system antically discuss and analyze which method is more suitable for the application of energy.

### 3.1. Sticking Method

Low-melting liquid metal will form an atomic oxide film attached to the surface of the liquid metal when attached in the air immediately. Common low melting liquid metals are gallium (Ga), indium (In), tin (Sn) or metal alloys such as gallium indium alloy (EGaIn), gallium indium tin alloy (EGaInSn) and so on. In air, since the Gibbs free energy of gallium oxide is relatively small than its alloys, it is more inclined to form a gallium oxide film on the metal surface [73,74]. At present most of the work focused on the liquid metal with low melting metals and alloys, the choice of the low melting point metal tends to Gibbs free to smaller metal oxide, when melting point to two kinds of metal in the molten state, the Gibbs free energy of the smaller metal oxide is preferred in the metal surface precipitation, then through the study of the pretreatment of base. Luckily, since the force between the substrate and the oxide film is larger than that between the liquid metal and the oxide film, the corresponding oxide film will be well obtained. Therefore, a simple sticking method can be used, and an oxide film at the atomic level can be obtained on the base by pre-processing the base.

In 2017, Ali Zavabeti et.al have first obtained extremely thin sub-nanometer layers of Ga_2_O_3_, HfO_2_, Al_2_O_3_, and Gd_2_O_3_ through room-temperature liquid metals as a reaction environment in Figure 2a [56]. Specifically, when the liquid metal alloy is prepared, the surface will soon form oxide when exposed to the air, and then the liquid metal oxide film can be obtained by pasting the liquid metal with the substrate. The as-prepared thin film shows the ultimate ultra-thin shape in TEM images as shown in Figure 2b. AFM characterization proves that each 2D nanosheets exhibits a super thin thickness of 2.8 nm, 0.6 nm, 1.1 nm and 0.5 nm, it is also proved that crystalline 2D material films can be obtained by metal melting corresponding to low Gibbs free energy metal oxides using this method. Subsequent articles have prepared other types of ultrathin 2D nanostructures by means of liquid metal synthesis. In 2009, Ali Zavabeti subsequently construct liquid metal Van der Waals (VdW) transfer method (refer to sticking method) is used to construct large-area heterostructures of atomically thin metal oxides of p-SnO/n-In_2_O_3_ with ease, which is shown in Figure 2c [71]. The liquid metal synthesis and transfer method therefore can potentially provide access to a variety of other heterostructures which are made of oxide layers that do not naturally exist as layered materials leading to interesting new discoveries. Further, high-resolution TEM (HRTEM) and selected area electron diffraction (SAED) pattern are all confirmed that well-fabricated In_2_O_3_ and SnO from liquid metal of Tin and Indium respectively in Figure 2d,e. In order to obtain ultrathin 2D nanomaterials with high crystallinity, it is necessary to remove the existing oxide film on the surface of the liquid metal precursor. In 2020, Kourosh Kalantar-Zadeh and his colleagues proposed a self-deposition of 2D molybdenum sulfide is shown by introducing a molybdenum precursor onto the surface of a eutectic alloy of gallium and indium (EGaIn) [72]. In detail, HCL is used to remove the oxide layer of the liquid metal, providing a proper environment for the reaction between the EGaIn and solvent, while the sufficient potential derived from EGaIn facilitated the deposition process obtaining a highly crystalline 2H-MoS_2_ as shown in Figure 2f. Figure 2g,h further confirmed the crystallinity of the as-prepared 2H-MoS_2_, specifically, the d-spacing of [1010] plane is 2.76 Å, and the d-spacing of [1120] plane is 1.59 Å, which is in agreement with the reported structure of 2H-MoS_2_ crystal, that work demonstrates a fundamentally new capability for the formation of large-scale 2D TMDs.

### 3.2. Gas Injection Method

Another method to obtain 2DMOs is the blowing method, also named as gas injection method, which is similar to the direct bonding method. The core of this method is as follows: When the oxygen bubbles contact with the liquid metal surface, the oxides on the liquid metal surface will precipitate out in the form of bubbles, and these oxides will appear in the supernatant of the solution due to low buoyancy. Compared with the sticking method, the gas injection method is the same as the bonding method, that is, the oxide type of the surface blow bonding is adjusted by adjusting the molten metal. The difference is that by adjusting the solvent type and reaction temperature, precise two-dimensional ultrathin nanomaterials (2DMOs) can be obtained with higher yields than the paste method.

Ali Zavabeti et al. first prepared Ga_2_O_3_ and HfO_2_ thin film via gas injection method applying deionized water as a solvent as shown in Figure 3a [56]. The thicknesses of the oxide films observed by AFM were 5.2 nm and 0.46 nm (Figure 3b,c), respectively. His first discovery provided new ideas and directions for the synthesis of other metal oxides. According to the interpretation of Gibbs free energy, metal oxides with smaller metal oxides must be selected for metal smelting in order to obtain pure metal oxides. Torben Daeneke et al. reported a liquid metal synthesis strategy [66], in which nanoscale titanium (Ti) powder was melted with liquid metal gallium as the raw material, and micron-scale ultrathin rutile TiO_2_ nanosheets with uniform thickness (20.5 nm) could be obtained by blowing air as shown in Figure 3d. HRTEM images and AFM characteristics further verify this point in Figure 3e,f.

Subsequently, Hu er et al. modified the previous blowing device, while designing a U-shaped device that could collect high-yield 2D metal oxide material films as shown in Figure 3g [52], so that the prepared nanomaterials can reach high productivity of 0.879 g. Figure 3h are the TEM images that shows great properties of large size and low thickness of the as-prepared nanoflakes. Afterward, energy dispersive spectroscopy (EDS) mapping of the nanoflake endows great uniformity and consistency which are shown in Figure 3i. In addition to this, the authors also found that part of the solvent is advantageous to the successful preparation of nanomaterials, this is due to the interaction of organic solvent adsorption oxide and metal oxide and liquid metal interaction force of competition as a result when organic solvents containing hydroxyl groups on the preparation of two-dimensional SnO_x_ effect is best, this is because the rich hydroxy solvent has good oxide affinity, It can better strip metal oxides from liquid metal surfaces. Their findings provide guidance for the synthesis of more diverse 2D oxides from liquid metals in non-aqueous dispersed systems.

### 3.3. Ultrasound Method

Uniform liquid metal nano-particles can be successfully prepared by ultrasonically in solvent with antioxidant capacity [63,64,65,68]. Notable, the inner core of the as-prepared nano-particles is liquid metal, the surface layer is the metal oxide, and the outermost layer is an antioxidant layer generated by the selected solvent. Generally, the selected solvent must meet several conditions, (1) Non-toxic and pollution-free. (2) Not participate in the reaction. (3) Can prevent the oxidation of nanoparticles after ultrasonic. Thus, organic solvents rich in hydroxyl and carboxyl groups are usually chosen. Under certain conditions, anhydrous solvents such as alcohol can also be utilized to disperse liquid metal. However, they suffer from poor stability and the fact that it is easy to reunite into liquid metal. To be notable, it is worth noting that the temperature during the whole ultrasound process should keep in an ice bath environment, which can effectively prevent the nanoparticles from oxidizing, or cause the nanoparticles to aggregate and re-form liquid metal.Ren and his colleagues utilized thiol and ethanolic solution as a solvent to produce EGaInSn (nanodroplet) NDs from EGaInSn bulk alloy under oscillating shear force due to the ultrasound process as shown in Figure 4a [69]. When EGaInSn is under the action of solvent ultrasound, the size of the metal droplets will gradually decrease, and an oxide layer will appear on the surface due to the participation of water molecules in the solution. It is worth noting that, as a result of the competition between the self-assembly of the thioligands and the oxidation process, Figure 4b shows a protective layer is formed outside the oxide layer to ensure good mechanical stability of the liquid metal nanoparticles and will not be merged into a liquid state when exposed to air. Figure 4c further illustrates that the average size of EGaInSn nanoparticles produced by 60 min ultrasound is about 110 nm, which proves that liquid metal can obtain homogeneous nanomaterials by ultrasonic method.

Subsequently, sodium alginate is an active substance extracted from alginic acid, which is widely used in food additives and biocompatible materials. Its biggest advantage is environmental compatibility and non-toxicity. Combined with liquid metal particles, as an antioxidant layer of metal oxide film, sodium alginate has been widely used in many medical and biological scenarios [67,68]. Li et al. reported a biocompatible aqueous solution of sodium alginate as a solvent to disperse liquid metal gain particles [67]. As a natural alginate, sodium alginate’s safety and biocompatibility are commendable, as shown in Figure 4d. It is worth noting that under the action of ultrasound, alginate promoted the formation process of EGaIn nanoparticles through the coordination between the carboxyl functional group and Ga ions. After 30 min of continuous ultrasound, the solution appeared opaque, and liquid metal particles with an average size of 80–250 nm were obtained after washing, as shown in Figure 4e. In addition, due to the formation of microgel shells around EGaIn droplets during the ultrasonic process of sodium alginate, the colloid and chemical stability of aqueous EGaIn nanodroplet ink is guaranteed due to the reduced oxygen permeability of the microgel shells. It is worth noting that the average diameter and oxide thickness of nanoparticles can be effectively adjusted by changing the concentration of sodium alginate. Figure 4f shows that with the increase of the concentration of sodium alginate solution, the thickness of the oxide film on its surface will become thicker. This is because carboxylic groups in sodium alginate will react with Ga^3+^ in the solution to generate eggshell-like microscopic gel products. However, when its thickness increases to 20 nm, it limits the surface diffusion of Ga^3+^. The same is true for its particle size. With the increase of concentration, its size first decreases and then increases, which is caused by the inhibition of oxidation by the microgel film.

## 4. Tow-Dimensional LM-Based 2DMOs Applications

### 4.1. CO_2_ Reduction

Nowadays, due to the situations that global warming and the limited storage of the greenhouse gases such as CO_2_, the great convert greenhouse gases into useful fuels and chemical products have become an appealing way to solve this problem and make the climate of the future stable [73,74,75,76,77]. It is widely known that solid metals perform excellent in electrocatalytic and photocatalytic of CO_2_ reduction, but there are still some obstacles that need to be solved such as the fact that carbonaceous materials will adhere to the catalytic sites and affect the reduction. What’s more, the metal oxides that have been examined for converting CO_2_ such as TiO_2_ [78], Cu_2_O [79] and Ag_2_O [80], are also active for electrocatalytic CO_2_ reduction. Adopting metal/metal oxide heterostructures is also an effective way for accomplishing better catalysis while the metal/metal interaction only performs satisfactorily in the liquid phase. Thus, the metal/oxide interactions have been widely utilized to improve the kinetics for chemical catalysis in gas phase.

CO_2_ is a stable molecule and thus designing a process that could convert at a low overpotential and room temperature is challenging. One of the methods to conquer this difficulty is to dissolve CO_2_ in a liquid environment to take an electrocatalytic. Based on the characteristic that the high electrical conductivity of the liquid metals which also allows the solid products could exfoliate immediately to keep the active sites of the interface accessible, thus using the liquid metal to reduce the CO_2_ becomes a promising choice [81,82]. Jumma Tang et al. exploited a suspension of Ga and AgF salt mixes as the precursors of the co-catalysts employed to stimulate the CO_2_ reduction which would get the best outcomes [81]. Dimethylformamide (DMF) owes high CO_2_ solubility and good stability under mechanical agitation, so they choose DMF as the solvent. In order to remove the native oxide on the surface of Ga, the solution always contained 0.1 M HCl. During the reduction process, they observed the CO_2_ molecules near the interface were reduced to form carbonaceous sheets due to the ultrasmooth nature of the LM. Due to the weak impact of Van der Waals force between the products and LM’s interface, making the LM is a kind of sustainable material to reduce CO_2_. The conversion provides a promising future of CO_2_ reduction. The carbonaceous materials are produced at low costs and can also be used as construction materials, such as supercapacitor ingredients, photoadsorbents or catalyst support. And because of the stable characteristic that the solid carbonaceous materials have, enabling a negative CO_2_ emission technology. To sum up, this novel concept will profoundly impact the future of carbon capture technologies and relevant industries.

Dorna Esrafilzadeh et al. demonstrated that a liquid metal electrocatalytic system containing cerium could convert the CO_2_ to carbonaceous and graphitic products at room temperature (Figure 5a) [57]. Compared to the other catalytic system based on liquid metal, due to the existence of the Ce, which could enhance the performance of the catalytic. HRTEM analysis of LMCe droplets (Figure 5b) revealed the formation of a 2D cerium oxide layer with a thickness of ~1.7 nm on the LM surface. They use gas chromatography detected the gas products including H_2_ and CO and through the faradaic efficiencies of the different potentials of LMCe3% for the production of CO, H_2_, and solid carbonaceous material which shows that the CO_2_ could convert into the solid products at a low onset potential (Figure 5c).

While most solvents become unstable at a high temperature in oxygen, molten salts are stable within a wide temperature window. In this regard, Zheng Hu et al. produced 2D oxide by bubbling oxygen through the liquid metal which used the molten salts as dispersion solvent for 2D oxides (Figure 5d) [83]. They also recorded the polarization curve to confirm that in the CO_2_-saturated electrolyte the high activity in CO_+_ reduction reaction over hydrogen evolution reaction (Figure 5e). The Faradaic efficiency of products at different applied potentials (from −0.87 to −1.47 V vs. RHE) shows that CO_2_ can be efficiently converted into formation within a wide potential window (Figure 5f).

### 4.2. Battery

In the past decades, the development of renewable energy mainly based on electric energy has attracted extensive attention from society, among which how to improve its performance and ensure its stability is still a difficult problem to be solved [84,85]. At present, the dendrite issue is the most important phenomenon affecting the efficiency of the battery anode, the core of which is the short circuit and disorder caused by an excessively long dendrite tip. As the development of liquid metal batteries rapidly advances, significant efforts have been made in sodium-beta alumina batteries (NBBs) having a liquid Na anode and a β″-Al_2_O_3_ solid electrolyte (BASE). What’s more, improving the wetting behavior of liquid Na on the surface of β″-Al_2_O_3_ can produce lower operating temperature NBBs which could also increase battery performance. Thus, Dana Jin et al. coated a few nanometer-sized Bi-metal islands deposited on the surface of β″-Al_2_O_3_ as shown in Figure 6a [86]. Figure 6b shows the voltage profiles of the initial cycle towards a different state of β″-Al_2_O_3_ including bare, Bi-coated and scratched Bi coated. The result implies that reversible electrochemical reactions occurred in both Bi-treated cells, even during the initial cycle.

Fangjie Mo et al. used finite element analyses to prove that the liquid metal Ga could alleviate the mechanical degradation because of the volume change during the All-solid-state lithium-ion batteries’ working [87]. They confirmed that liquid Ga could wet the interface of Sb/LiBH_4_, it can accelerate the diffusion of electrons and ions, promoting the electrochemical performance of metal Sb. To further verify the detailed reaction pathway and the existence of liquid Ga of the GaSb anode, they employed the ex-situ TEM analysis to characterize the phase and structure at different states. Figure 6c exhibits SAED and high resolution (HR) TEM of the GaSb electrode discharged to 0.6 V and shows that Sb is fully litigated and Ga is partially lithiated, when the battery is charged back to 1.2 V, only Sb and Ga can be observed in the SAED and FFT images. The corresponding differential capacity (dQ/dV) curve of GaSb and TiS_2_ in half cell based on LiBH4 electrolyte are shown in Figure 6d.

### 4.3. Photodegradation

With the increasing global concern for environmental remediation, the development and application of various photocatalysts have aroused great interest in the decomposition of various organic pollutants [88,89]. However, the inefficient absorption of the solar spectrum, coupled with the rapid deactivation of photocatalysts, limits the practical application of many photocatalysts. In liquid metal oxide films, oxidation grafting is an excellent catalyst material, it has a good reduction potential as a wide band gap list of semiconductor materials, and it has five different, phase structures, setting him apart from the rest of the material [90,91]. However, as a wide-band-gap List of semiconductor materials, reducing its band-gap and extending its spectral absorption range to the visible band have far-reaching and positive implications. At present, there are many methods to synthesize oxide nanomaterials, including hydrothermal method and solvent method. Recently, the method of obtaining oxide nanomaterials by liquid metal has been accepted by more and more people. It should be noted that the oxide film obtained by liquid metal can introduce a defect state, which may lead to the change of the band gap of the oxide film, and the result is for photodegradation, it can control the broad spectrum absorption range of the material and increase its absorption ability of visible light.

Kalantar-Zadeh and his colleagues applied GaO_x_ nanoflakes as a fascinating photocatalytic material to decompose organic model dyes-Congo red (CR) [92]. It is noted that during the self-limiting process via the liquid metal-gallium, the bandgap of GaO_x_ could be well adjusted due to the trap states of liquid metal, leading to a satisfying fact that there is a large increase in spectral absorption range. Specifically, Figure 7a is the nanosheet material stripped from liquid metal. After annealing, the α-Ga_2_O_3_ nanosheet is obtained. It can be clearly seen that its size is about 250 nm and 500 nm. Subsequently, Figure 7b shows the changing trend of an absorption spectrum of the mixed solution of α-Ga_2_O_3_ nanoparticle catalyst and reactant CR during the photocatalytic reaction. The characteristic absorption peak of 500 nm showed a decreasing trend within 120 min, and the color of CR solution was degraded from red to pink to a colorless solution indicating the high catalytic efficiency of α-Ga_2_O_3_ nanoparticles for CR. Figure 7c further demonstrated the change in CR relative absorbance (A_t_/A_0_) and dye degradation rate, where 70% of CR was observed to be eliminated after 60 min in the presence of catalyst and light. The experimental results show that by using the self-limiting layer oxide film obtained during the self-oxidation of liquid metal and annealing it, effective solar energy GaO_x_ photocatalyst can be produced with high yield following a simple and cheap synthesis route.

In order to improve the photocatalytic performance of GaO_x_ material, the band gap of GaO_x_ material was effectively regulated. Later, Kalantar-Zadeh research group continued to in-depth research progress of GaO_x_ [93], they will be the sol thermal synthesis of Ga_2_O_3_ nanoparticles and liquid metal nanoparticles constitute a liquid metal/metal oxide (LM/MO) framework. The framework is based on the plasma effect. The effective introduction of Ga_2_O_3_ nanoparticles can form an ohmic contact with the liquid metal oxide film, and then reduce the potential barrier of GaO_x_ itself. Smaller photon energy can excite more photogenerated carriers for efficient photocatalysts. Specifically, as shown in Figure 7d, the LM/MO framework acts as an efficient ion and electron conversion window, endowing GaO_x_ nanoparticles with high redox potential, inhibiting their own electron-hole pair recombination, and broadening their spectral absorption range. The coupling effect between the two for efficient photocatalytic materials provides a new idea and thinking. Next, the photocatalytic properties of Ga_2_O_3_ nanoparticles, LM/MO skeleton, and their mixed products in different proportions were simulated by an xenon lamp light source. The test results show that the degradation efficiency of the original Ga_2_O_3_ nanoparticles is the lowest, which is due to the high bandgap of the material itself, and only has catalytic effects on ultraviolet light. This is followed by the slightly higher catalytic performance of the LM/MO skeleton due to the display of tunable plasmonic resonances capable of producing high sensitivity to low concentrations of heavy metal ions and enhanced solar driven photocatalytic activity. Next, the photocatalytic properties of Ga_2_O_3_ nanoparticles, LM/MO skeleton, and their mixed products in different proportions were simulated by xenon lamp light source. The test results show that the degradation efficiency of the original Ga_2_O_3_ nanoparticles is the lowest, which is due to the high bandgap of the material itself and only has a catalytic effect on ultraviolet light. Figure 6e shows when Ga_2_O_3_ nanoparticles cooperate with LM/MO skeleton, the degradation efficiency is greatly improved, and the proportion of 1wt % is the highest, which may be related to the consumption of too many holes in LM/MO skeleton. Furthermore, Figure 6f shows the photocatalytic stability diagram of adding 1wt % Ga_2_O_3_ nanoparticle to LM/MO skeleton. In the last four stability cycle diagrams, the photocatalytic efficiency does not decrease significantly, indicating its good catalytic efficiency and stability.

In summary, more and more efficient photocatalytic degradation agents based on liquid metal oxide films have been proposed, which are essentially liquid metal self-limiting layers.

## 5. Summary and Perspectives

In summary, the exploration of 2D liquid metal oxide semiconductors with unique advantages has motivated our subsequent research in the field of their energy storage applications. 2DMOs with sufficiently thin and large nanostructures and a variety of tunable bandgaps after modification are ideal materials to achieve high photoelectric conversion efficiency for new energy technologies. In this review, the latest research progress of two-dimensional metal oxide nanomaterials based on liquid metals as a new energy source is reviewed. Firstly, the surface oxidation process and in situ electrical replacement reaction of liquid metal are analyzed, which provides a theoretical basis for the realization of functional two-dimensional nanomaterials. Then, from the sticking method, gas injection method and ultrasonic method, the preparation methods of two-dimensional liquid metal oxides are briefly discussed. More importantly, we review the energy applications of 2D liquid metal oxides, including photocatalytic degradation, carbon dioxide emission reduction, and energy-related applications such as batteries, showing that they have made the above great progress and show clear advantages in new energy sources. Although 2DMOs have made remarkable progress in the application of new energy sources, they are still in an immature stage and require further design to face the future theoretical and practical challenges.

## Figures and Tables

**Figure 1 molecules-28-00524-f001:**
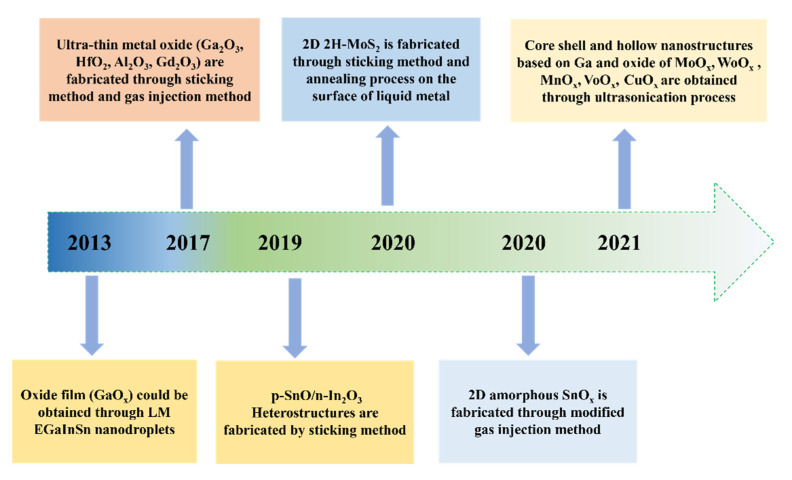
Timeline of Landmark Academic Works About 2DMOs from LM.

**Figure 2 molecules-28-00524-f002:**
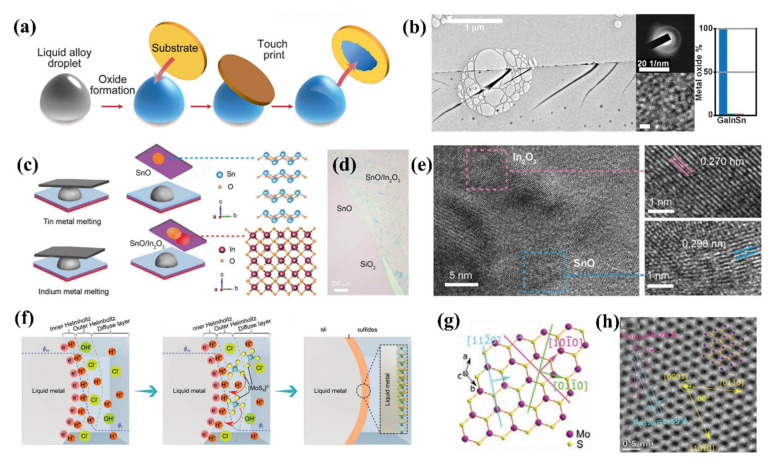
(**a**) Schematic diagram of the sticking method: Applying modified substrate to obtain liquid metal oxide film from liquid metal (**b**) TEM of Ga_2_O_3_ thin film and its corresponding SAED and HRTEM images. Reprinted with permission from Ref. [56] Copyright 2017 Science. (**c**) Schematic illustration of the 2D VdW transfer technique of SnO/In_2_O_3_ thin films from liquid metal (**d**) Optical image of the 2D SnO/In_2_O_3_ heterostructure (**e**) HRTEM image showing a highly crystalline structure with continuous lattice fringes across the SnO/In_2_O_3_ heterostructure interface. Reprinted with permission from Ref. [71] Copyright 2019 Advanced Materials Interfaces. (**f**) Schematic illustration of MoS_x_ formation on EGaIn (**g**,**h**) The crystal structure of MoS_2_ sheets. Reprinted with permission from Ref. [72] Copyright 2020 Advanced functional materials.

**Figure 3 molecules-28-00524-f003:**
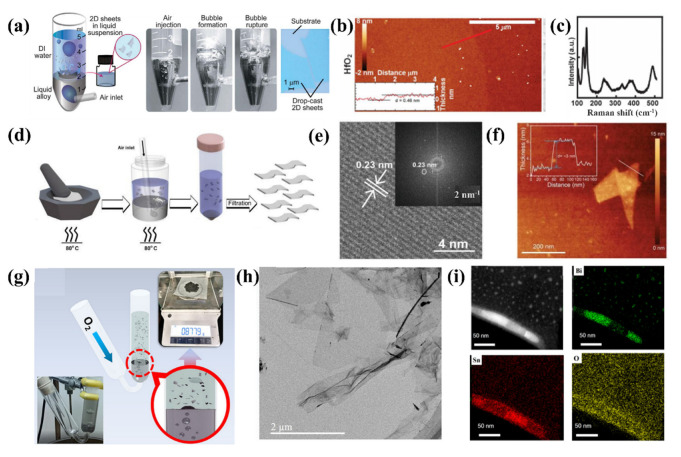
(**a**) Schematic representation of the gas injection method and the optical image of the obtained nanosheets (**b**) AFM and (**c**) Raman spectrum of the HfO_2_ nanosheets via gas injection method. Reprinted with permission from Ref. [56] Copyright 2017 Science. (**d**) Schematic representation of the gas injection method to prepare 2D TiO_2_ nanosheets (**e**) HRTEM images of the prepared 2D TiO_2_ nanosheets (**f**) AFM image of the prepared 2D TiO_2_ nanosheets. Reprinted with permission from Ref. [66] Copyright 2020 Chemical Communications. (**g**) The Schematic representation of injection of O_2_ into liquid metals capped with specific dispersion solvent to obtain SnO_x_ nanoflakes (**h**) TEM images of the prepared large SnO_x_ nanoflakes (**i**) STEM image and EDS mapping of the Bi, Sn and O edge of a nanoflake. Reprinted with permission from Ref. [52] Copyright 2020 Nano Letters.

**Figure 4 molecules-28-00524-f004:**
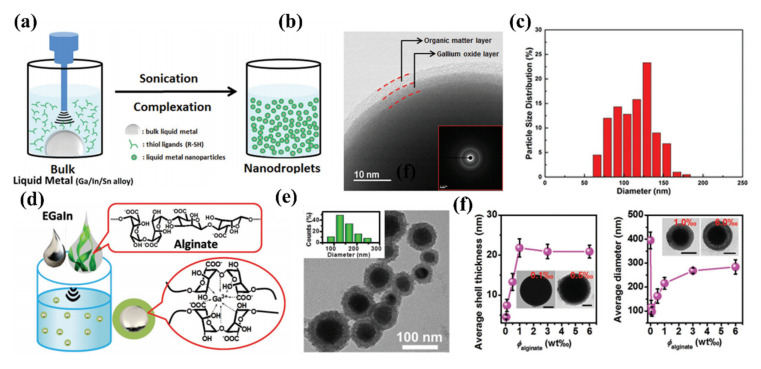
(**a**) Schematic illustration of the preparation route for EGaInSn nanodroplets (**b**) HRTEM image demonstrates the core–shell structure of EGaInSn nanodroplets, the inset is the corresponding SAED pattern (**c**) Size distribution of EGaInSn nanodroplets. Reprinted with permission from Ref. [69] Copyright 2016 Advanced functional materials. (**d**) Production of aqueous ink of EGaIn nanodroplets using alginate microgel as the solution (**e**) TEM image and diameter histogram of EGaIn nanodroplets (**f**) average shell thickness and average diameter of EGaIn droplets on alginate concentration. Reprinted with permission from Ref. [67] Copyright 2018 Advanced functional materials.

**Figure 5 molecules-28-00524-f005:**
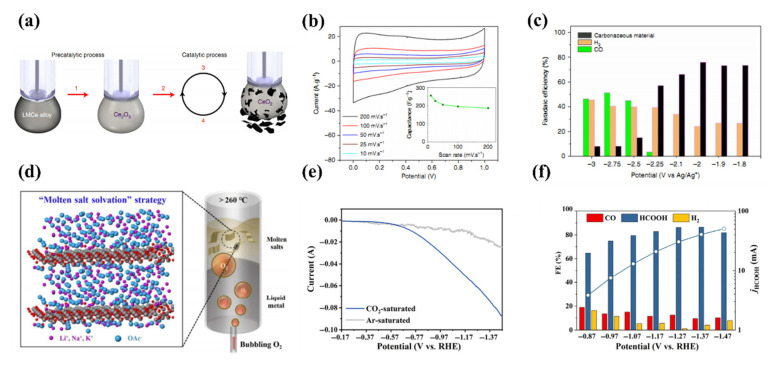
(**a**) Schematic of the catalytic process. The proposed process is based on operando Raman measurements, it includes pre-catalytic reactions and the catalytic cycle for the CO_2_ reduction to amorphous carbon sheets (**b**) Supercapacitor behaviour of the as-prepared materials, insert image are the calculated specific capacitance of the capacitor at various scan rates (**c**) Faradaic efficiencies of LMCe3% for the production of CO, H_2_, and solid carbonaceous material at corresponding potentials measured in CO_2_ saturated electrolytes. Reprinted with permission from Ref. [57] Copyright 2019 Nature Communications. (**d**) Principle for the growth and molten-salt assisted dispersion of 2D nanosheets from the liquid metal (**e**) Linear sweep voltammetry curve of SnO_x_ nanosheets catalyst in CO_2_-saturated 0.5 M KHCO_3_ aqueous solution (scan rate: 10 mV·s^–1^) (**f**) Faradaic efficiency of H_2_, CO and formate in the electrocatalytic CO_2_ RR with different potentials applied to the catalyst-loaded working electrode. Reprinted with permission from Ref. [83] Copyright 2021 Nano Research.

**Figure 6 molecules-28-00524-f006:**
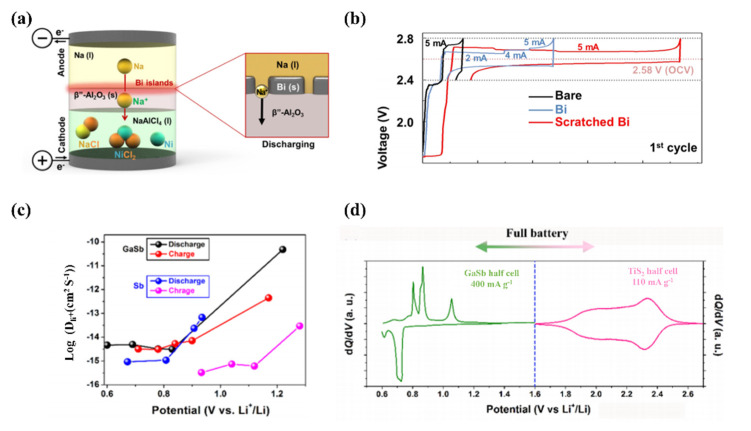
(**a**) Schematic of the Na/NiCl2 cell (**b**) Voltage profiles of the initial cycle of bare, Bi-coated, and scratched Bi-coated β″-Al_2_O_3_ cells. Reprinted with permission from Ref. [86] Copyright 2019 ACS Applied Materials & Interfaces. (**c**) Li-ion diffusion coefficients for GaSb and Sb electrodes at selected states (**d**) Typical dQ/dV curves of GaSb and TiS_2_ electrodes. Reprinted with permission from Ref. [87] Copyright 2020 ACS Applied Materials & Interfaces.

**Figure 7 molecules-28-00524-f007:**
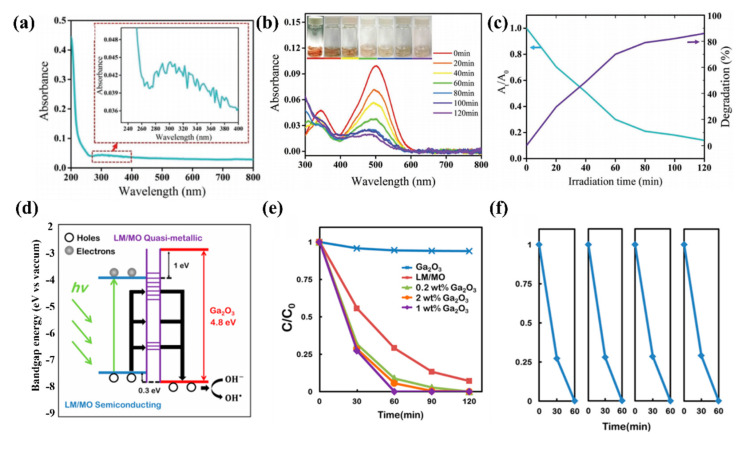
(**a**) The nanosheet material stripped from liquid metal (**b**) The changing trend of the absorption spectrum and CR’s degradation process (**c**) CR relative absorbance (A_t_/A_0_) and dye degradation rate. Reprinted with permission from Ref. [92] Copyright 2017 Advanced functional materials. (**d**) Band structure diagram of the combined system (**e**) Degradation of 10 µM CR in the presence of LM/MO frameworks incorporated with different loadings of Ga_2_O_3_ nanoparticles (**f**) Degradation of 10 µM CR in the presence of one sample of LM/MO frameworks with 1 wt % incorporated Ga_2_O_3_ in four consecutive cycles. Reprinted with permission from Ref. [93] Copyright 2015 ACS Applied Materials & Interfaces.

**Table 1 molecules-28-00524-t001:** Key factors of 2DMOs obtained from liquid metal.

Method	Liquid Metal	Oxide Film	Condition	Substrate (Solvent)	ΔG (kJ mol^−1^)	Crystallization	Thickness (nm)	Ref.
Sticking method	EGaInSn	Ga_2_O_3_	Room temperature; containing less than 0.1% of oxygen	Si, SiO_2_, Pt coated Si wafers, Au coated Si wafers, quartz, glass, ITO	−998.3	No	2.78	[56]
	EGaInSn	HfO_2_			−1088.2	Yes	0.64	
	EGaInSn	Al_2_O_3_			−1582.3	Yes	1.1	
	EGaInSn	Gd_2_O_3_			−1732.3	Yes	0.51	
	Sn	SnO	Oxygen Concentration (10–100 ppm)	SiO_2_/Si SiO_2_/Si	−251.9	Yes	1	[71]
	In	In_2_O_3_			−830.7	Yes	4.5	
	EGaIn	MoS_2_	Room temperature	SiO_2_/Si/sapphire	n/a	Yes	0.7	[72]
Gas injection method	EGaInSn	Ga_2_O_3_	Room temperature;	Distilled water	−998.3	No	5.2	[56]
	EGaInSn	HfO_2_			−1088.2	No	0.46	
	Ga	TiO_2_	80 °C heating	Distilled water	−888.8	Yes	3	[66]
	Sn	SnO_x_	Oil bath at 180 °C with the bubbling of oxygen gas at a rate of 100 sccm	Diethylene glycol	−251.9	No	5	[52]
Ultrasound method	EGaIn	Ga_2_O_3_	Ice-water bath	Aqueous alginate solution	−998.3	No	0.5–3	[67]
	EGaInSn	Ga_2_O_3_	60 min at a cold water bath at about 20 °C	ethyl 3-mercaptopropionate	−998.3	No	2–3	[69]
	EGaInSnZn	MoO_x_	10 min at a cold water bath at about 20 °C	Na_2_MoO_4_ ·2H_2_O	−465.1	No	n/a	[70]
		WoO_x_		Na_2_WO_4_ ·2H_2_O	−533.9	No	n/a	
		MnO_x_		KMnO_4_ ·2H_2_O	−533	No	n/a	
		VoO_x_		Na_3_VO_4_ ·2H_2_O	−1139.3	No	n/a	
		CuO_x_		Cu(NO_3_)^2^ ·2H_2_O	−129.7	No	n/a	

## Data Availability

Not applicable.

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
