# Peer review of "Synthesis and Application of Liquid Metal Based-2D Nanomaterials: A Perspective View for Sustainable Energy"

_molecules, 2023, doi:10.3390/molecules28020524_

Round 1

Reviewer 1 Report

In the review with title “ Synthesis and Application of Liquid metal based-2D Nano- 2 materials: A Perspective View for Sustainable Energy by Gengcheng Liao et al. In this review, the authors presented some methods of synthesis of liquid metal- 2D Nanomaterials and some applications. In addition, the authors have presented some information about the properties of liquid metals and surface oxidation. Overall, the work may be of interest to a broader audience and is very important. However, the authors should address the following points outlined below to improve the scientific quality. After the suggested revisions are carefully addressed, this work may be considered for publication in Molecules Journal.

1-     Introduction section page 2 lines 59-62 : “Specifically, low melting point liquid metals, such as pure Ga, EGaIn, EGaInSn et al. will form  self-limiting oxide films in air at room temperature, which is considered a natural two- dimensional semiconductor. Please rewrite the sentence and write the complete name of the abbreviations for example eutectic gallium–indium (EGaIn).

2-     Introduction page 2 lines 85-86 change light degradation to photodegradation

3-     Page 3 lines 100-101, In the absence of oxygen, Ga appears in Ga0 state, but at room temperature, Ga 0 and Ga3+ coexist on the surface of gallium.( please check this sentence).

4-     Page 3 line 104, 108, 109 correct the symbol SN to Sn.

5-     Rewrite the sentence in page 3 lines 108-110 “And that's because SN is more likely to be attached to an oxygen group, and that has a lot to do with the valence electrons of SN such as the +2 and +4 valence electrons.

6-     Page 4 lines 153 to lines 164 please rewrite these sentences.

7-     Page 4 lines lines 167-171 “At present most of the work will be liquid metal with low melting point  metal and alloy, the choice of the low melting point metal tend to Gibbs free to smaller  metal oxide, when melting point to two kinds of metal in the molten state, the Gibbs free  energy of the smaller metal oxide is preferred in the metal surface precipitation, then through the study of the pretreatment of base. This sentence is so long and nor clear.

8-     The title “Van der Waals paste method” Please correct this title to be more scientifically. In addition provide analysis of the processing parameters of this method.

9-     Delete the sentence lines 215-216 page 5 “It comes into contact with molten liquid  metal via oxygen”.

10- The sentence in the page 5 lines 218-220 “Compared with the direct bonding method, the air blow bonding method is the same as the bonding method, that is, the oxide type of the surface blow bonding is adjusted by adjusting the molten metal.  Rewrite this sentence in clear form and avoid repeating.

11- The sentence “According to the interpretation of Gibbs free energy, metal oxides with smaller metal oxides must be selected for metal smelting in order to obtain pure metal oxides. Not clear, please rewrite it.

12- Page 5 line 30, please correct the symbol : ti  and also retype the sentence.

13- Page 6 line 240 what do you mean by “ part of the solvent”.

14- Regarding the ultrasound method for synthesis of liquid metal- 2D Nanomaterials could the authors provide analysis in tables for the various parameters affecting the formation of liquid metal- 2D Nanomaterials.

15- Please avoid repeating the information. For example:   Page 11 lines 452-452 “The test results show that the degradation efficiency of the original Ga2O3 nanoparticles is the lowest, which is due to the high bandgap of the material itself, and only has catalytic effect  on ultraviolet light.  The author in the same paragraph lines 460- 462 “The test results show that the degradation efficiency of the original Ga2O3 nanoparticles is the lowest, which is due to the high bandgap of the material itself and only has a catalytic effect on ultraviolet light.

Reviewer 2 Report

Liao et al. review the research field of liquid metal based 2D nanomaterials in terms of synthesis as well as in terms of applications.  The manuscript is prepared with care, many important recent papers in the field are reviewed and overall the work is considered as suitable for publication in Molecules however, after some important modifications which I list below:

a) The abstract needs to be revised. There are various long sentences which fail to convey the message to the reader efficiently.  I suggest that the authors carefully revise the abstract in terms of language as well as in terms of scientific accuracy. 

b) In section 2, properties of LM 2D nanomaterials,  I would encourage the authors to include tables summarising the most important properties of the corresponding materials listing the relevant citations.  This is a more effective way of reviewing this part.

c) The introduction requires some modifications. The importance of the present review is not highlighted. Why this field of research and technology is /could be important is not fully justified. Parts of the introduction need to be revised.  Also, a schematic illustration  indicating the evolution of the field is considered as essential. 

Round 2

Reviewer 1 Report

Accept after Moderate English changes are required.